# Inhibition of Satellite RNA Associated Cucumber Mosaic Virus Infection by Essential Oil of *Micromeria croatica* (Pers.) Schott

**DOI:** 10.3390/molecules24071342

**Published:** 2019-04-05

**Authors:** Elma Vuko, Gordana Rusak, Valerija Dunkić, Dario Kremer, Ivan Kosalec, Biljana Rađa, Nada Bezić

**Affiliations:** 1Faculty of Science, University of Split, Ruđera Boškovića 33, HR-21000 Split, Croatia; dunkic@pmfst.hr (V.D.); radja@pmfst.hr (B.R.); bezic@pmfst.hr (N.B.); 2Faculty of Science, Department of Biology, University of Zagreb, Roosveltov trg 6, HR-10000 Zagreb, Croatia; gordana.rusak@biol.pmf.hr; 3Faculty of Pharmacy and Biochemistry, University of Zagreb, A. Kovačića 1, HR-10000 Zagreb, Croatia; dkremer@pharma.hr (D.K.); ikosalec@pharma.hr (I.K.)

**Keywords:** *Micromeria croatica*, CMVsat, essential oil, β-caryophyllene, caryophyllene oxide

## Abstract

The present results dealing with the antiphytoviral activity of essential oil indicate that these plant metabolites can trigger a response to viral infection. The essential oil from *Micromeria croatica* and the main oil components β-caryophyllene and caryophyllene oxide were tested for antiphytoviral activity on plants infected with satellite RNA associated cucumber mosaic virus. Simultaneous inoculation of virus with essential oil or with the dominant components of oil, and the treatment of plants prior to virus inoculation, resulted in a reduction of virus infection in the local and systemic host plants. Treatment with essential oil changed the level of alternative oxidase gene expression in infected Arabidopsis plants indicating a connection between the essential oil treatment, *aox* gene expression and the development of viral infection.

## 1. Introduction

Essential oils (EOs) have become a subject of interest due to their numerous biological effects, attributed primarily to prooxidant effects at the cellular level [1,2]. Significant quantitative and qualitative variations of essential oil components have been observed within and between species, but also within populations and even within individuals, with strong implications for plant–herbivore and plant–pathogen interactions. The defensive action of essential oils can be direct, through toxicity, while some volatile organic compounds released upon chewing act as specific cues to attract parasitoids [3,4,5]. While many individual compounds are toxic to herbivores, others act as antifungal, antibacterial or allelopathic agents, or for priming of systemic defences in both the host and neighbouring plants [6,7,8,9]. A recent area of research is the antiphytoviral activity of EOs. Although limited, the results to date indicate that these plant metabolites can trigger a response to viral infection [10,11,12,13,14,15,16]. Qualitative differences in EO composition of different plant species do not exhibit a uniform response that would explain the mechanism of antiphytoviral activity. Previous studies on the EOs of different plant species have led to the conclusion that sesquiterpene-rich oils can reduce viral infection in local host plants [10,12,13,16]. Synthesis of certain *Arabidopsis thaliana* sesquiterpenes, including β-caryophyllene, are induced by the phytohormones gibberellin (GA) and jasmonate (JA), indicating that the production of these sesquiterpenes is integrated within the GA and JA signalling pathways [17]. Sesquiterpenes are the main oil components of the EO of *Micromeria croatica* (Pers.) Schott described by Kremer et al. (2012) [18], and this study aimed to determine its antiphytoviral potential for the prevention of diseases caused by plant viruses. As endemic Illyric-Balcanic species with small isolated populations, *M. croatica* (Lamiaceae) has been explored insufficiently to date. The EO composition of certain species can display annual variations, and variations in composition between populations. Genetic factors, evolution, geographic variation, environmental conditions (i.e., harvest date, planting time), physiological variations (i.e., organ and leaf position), and developmental stage are known to affect the biosynthesis of EOs [19,20,21]. Thus, the first objective of this study was to analyse the chemical composition of the EO of *M. croatica* collected at three localities on Mt Velebit (Croatia), and to establish annual variations in oil composition, comparing the present results with those of a previous study [18]. The second objective was to analyse the uniformity of oil composition between *M. croatica* populations and to statistically analyse the results to clarify variations in the EO profile. The effect of these variations in the EO profile on the antiphytoviral activity of oil of *M. croatica* was examined to determine whether this activity is independent of oil composition. Cucumber mosaic virus (CMV) was selected for determining the antiphytoviral activity of EO due to its wide host range of some one thousand plant species, including many of the most important crops. Some strains of CMV are associated with satellite RNA (satRNA), which depends on their helper virus for replication and movement [22] and can have pronounced effects on host plant symptoms [23].

With the exception of our study dealing with the antiphytoviral activity of the EO of *Micromeria graeca* [16], no other literature reports on the antiphytoviral activity of EOs within the genus *Micromeria* were available. In this sense, the aim of this study was to provide answers to the following questions: (i) do the EO and the dominant components of oil (DC) of *M. croatica* have antiphytoviral activity; (ii) do different application methods of EO/DC affect antiviral activity in local host plants; (iii) does an extended incubation time of the virus with EO/DC reflect on the antiviral activity in local host plants; (iv) do EO/DCs have the ability to activate a defensive response of local host plants against viruses; (v) does the prolongation of treatment with EO/DC prior to inoculation reflect on the antiviral activity in local host plants; (vi) do the above treatments affect virus concentrations in systemic host plants; (vii) is there a correlation between the antiviral activity of the EO and the antiviral activity of the DC of oil; and finally, (viii) is the antiviral activity constant due to differences in oil composition?

The next step was to establish the antiviral activity of EOs by testing the expression of one of the genes important in plant defence against pathogens. Alternative oxidase (AOX) is the terminal oxidase of the cyanide-resistant alternative respiratory pathway in plants and has been implicated in resistance to viruses [24]. While AOX abundance and alternative pathway activity are low in unstressed plants, both increase when plants are subjected to suboptimal conditions, such as chilling [25] or pathogen attack [26]. The AOX limits the formation of mitochondrial reactive oxygen species (ROS) in all plant species [27], and in stress conditions, AOX reduces ROS formation and has an active role in the adaptation of Arabidopsis to cold conditions [28]. Different species under similar stress conditions have a different response to the synthesis of AOX, though several Arabidopsis studies have described *Aox1a* as the most stress responsive *Aox* gene [29]. Based on the above, we proposed that at least one part of defence response mediated by EO involves a change of expression of the *Aox* gene. Since CMV is able to infect *Arabidopsis thaliana*, we took advantage of this model system to investigate the correlation between essential oil treatment, *Aox* gene expression and the development of viral infection.

New EO/virus/host combinations and methodological approaches in this field enabled us to correlate EO composition/method of application with antiviral activity-gene expression. Although no single answer is expected to explain the antiphytoviral activity of EOs, answers to the above questions will improve the existing knowledge about the antiphytoviral activity of EOs, and our understanding of the role of EOs in plant-virus interactions.

## 2. Results and Discussion

### 2.1. Essential Oil Composition

The EO of the endemic *Micromeria croatica* from Mt. Velebit, Croatia (localities Bojinac (Boj), Bačić kuk (Bk), and Stupačinovo (St)) was subjected to detailed GC and GC-MS analysis to determine chemical composition and the similarity and variability in oil composition depending on locality. The chemical composition of EOs, the percentage of identified components and their retention indices are given in Table 1, where all the components are arranged in order of their elution from the VF-5ms capillary column [30]. The overall composition of EO of *M. croatica* is characterized by a high percentage of sesquiterpenes and monoterpenes (Table 1). In the oil from Boj, sesquiterpene β-caryophyllene (25.2%) is the dominant component, followed by oxygenated sesquiterpene caryophyllene oxide (10.1%). In the oil from Bk, the cumulative amount of monoterpenes is substantially higher than sesquiterpenes, although the oxygenated sesquiterpene caryophyllene oxide (21.1%) is a major component of this oil. In the oil from St, caryophyllene oxide (20.2%) is the major compound, followed by β-caryophyllene (10.2%) (Table 1). Minor qualitative and quantitative differences in the annual oil composition at the localities Boj and Bk were observed in comparison to previous results presented by Kremer et al. (2012), though the major components in oil composition remained unchanged [18]. Namely, β-caryophyllene and caryophyllene oxide were also identified as the dominant components of oil at the described localities [18]. Due to their high content in the composition of oil, β-caryophyllene and caryophyllene oxide were also subjected to n antiphytoviral assay to investigate their role in antiviral activity of the oil. The composition of oil at the St locality (Table 1) presents new data and consequently was not comparable to previous results at the annual level. Principal component analysis (PCA) was employed to provide an overview of the capacity to distinguish essential oil components based on GC-MS data (Figure 1). The common components of EO composition at the localities Boj, Bk and St clustered in the middle of the diagram. The EO components grouped at the periphery of the diagram show qualitative discrepancies in oil composition among the localities (Figure 1). Based on Figure 1, the EO composition in the studied populations of *M. croatica* can be considered to be quite uniform. A somewhat more pronounced degree of specificity was observed in the composition of oil from St (Figure 1). Cluster analysis classifies the number of samples studied into groups according to the chemical composition of essential oil by ‘magnifying’ their similarities. Multivariate analysis of the samples differentiated two well-defined clusters of *M. croatica* (Figure 2), in which one cluster included the localities Boj and Bk, and the locality St was isolated in its own cluster, indicating the specificity of its EO composition.

### 2.2. Antiphytoviral Activity

In the field of plant virology, there is a limited amount of information on the effects of EOs on viruses or viral infections in plant systems [10,11,12,13,14,15,16]. The first objective of this part of the study was to test the hypothesis on the antiviral potential of the EO of *M. croatica.* Thus, the EO/DC of oil was rubbed onto leaves of local and systemic host plants simultaneously with virus inocula, or the EO/DC of oil were introduced into host plants prior to virus inoculation. The aim was to gain insight into the antiphytoviral activity of the oil by providing data on EO-plant-virus interactions using a variety of inoculation methods of local and systemic host plants. In simultaneous inoculation on the leaves of local host plants, the virus was incubated with EO/DC of oil for 0, 1, 2 and 3 h prior to inoculation. When inoculation was carried out after a short incubation period (0 h), EO Boj and β-caryophyllene were most effective in reducing the number of local lesions compared to the control (33.9% and 38.3%, respectively) (Figure 3). EO St and caryophyllene oxide were most effective during the experiment with two-hour preincubation with the virus in comparison to the control (34.5% and 33.4%, respectively), while EO Bk was the most effective after one-hour preincubation (33.6%) (Figure 3). Three-hour preincubation was statistically ineffective for reduction of local lesions in all experimental groups, with the exception of β-caryophyllene, which inhibited lesions by 25.3% (Figure 3). Based on these results, Pearson’s correlation coefficient established a positive correlation of the antiviral activity of EO Boj with β-caryophyllene. Antiviral activity of EO Bk and EO St positively correlated with the antiviral activity of caryophyllene oxide.

The next aim of this study was to determine whether different methods of EO/DC application affect antiviral activity in local host plants and whether EO/DC has the ability to activate the defensive response of local host plants against viruses. Thus, local host plants were treated with EO/DC for one, two and three consecutive days prior to virus inoculation. Although all pretreatments significantly reduced the number of local lesions, the strongest antiviral effect was manifested after three days of treatment in all experimental groups. Reduction of local lesion numbers ranged between 66.8% and 71.4% (Figure 4), confirming the hypothesis of the antiviral activity of EO of *M. croatica*. Pearson’s correlation coefficient showed a positive correlation of the antiviral effect of EOs with both dominant components of oil. This part of the study led to the conclusion that simultaneous inoculation of virus inocula and EO/DC of oil is less effective for the reduction of local infection than treatment prior to inoculation (Figure 3, Figure 4). Furthermore, preincubation of viral inoculum with EO/DC that lasted more than two hours did not enhance the antiphytoviral effect (Figure 3). Due to the volatility of EOs, the concentration of active compounds was likely reduced. On the contrary, longer pretreatment of host plants with EO/DC of oil prior to inoculation enhanced the antiphytoviral effect (Figure 4). Based on the above, it can be concluded that induction of the defence response of local host plants is the mode of antiviral activity of EO of *M. croatica*, as opposed to direct “in vitro” inactivation of the viral particles.

Based on this promising antiviral activity of EO of *M. croatica* in local host plants, several additional parameters were examined to further clarify its antiviral activity. According to our knowledge, the effect of EOs on virus replication in systemic host plant has not yet been tested. With the aim of determining whether the above treatments affect the spread of the virus in systemic host plants, virus concentrations were compared in upper, new-grown leaves of EO-treated and untreated *Nicotiana megalosiphon* plants continuously over seven days post inoculation (p.i.). In that sense, systemic host plants were: 1. simultaneously inoculated with virus and EO/DC of oil; 2. treated with EO/DC three days prior to virus inoculation; or 3. continuously treated with EO/DC three days prior and seven days after virus inoculation. The virus concentration was determined in systemically infected upper leaves collected at 1, 3, 4 and 7 days p.i. to provide complete information on viral dynamics possible influenced by the EO.

In simultaneous inoculation with the virus, both EO Boj and β-caryophyllene reduced virus concentrations on day 1 p.i. by 23.8% and 25.0%, respectively (Figure 5). Inhibition of virus concentration had a decreasing tendency in later stages of infection in all experimental groups in this experiment (Figure 5). A positive correlation of the antiviral effect of EO Boj with β-caryophyllene was established by Pearson’s correlation coefficient. Pretreatment with EO/DC of oil was more successful in reducing virus concentration than simultaneous inoculation in all experimental groups. Namely, EO Boj and β-caryophyllene reduced virus concentrations on day 1 p.i. by 26.3% and 21.8%, respectively, EO Bk and caryophyllene oxide were the most effective on day 4 p.i. with inhibition of 28.3% and 28.6%, respectively, while pretreatment with EO St reduced virus concentrations on day 3 p.i. by 25.0% (Figure 5). Pearson’s correlation coefficient established a positive correlation of the antiviral activity of EO Bk with caryophyllene oxide. Continuous treatment of systemic host plants with EO/DC prior to and after inoculation did not improve the antiphytoviral effect over treatment prior to inoculation (Figure 5). Based on the overall results of inhibition of viral infection by EO, it can be concluded that the antiphytoviral activity of *M. croatica* EO in local host plants positively correlated with the activity of the main oil components. The observed discrepancies from this rule in the systemic host plants provide the answer to the question of whether antiviral activity is constant given the differences in oil composition. Since both the PCA and cluster analysis showed that composition of the oil from locality St was specific (Figure 1, Figure 2), it is possible that this slightly different oil composition affects its antiviral activity. It is possible that aside from the main components, other components in this oil or their synergistic effects contributed to the antiviral activity of oil from St. A similar conclusion was reported in previous studies [10,16] and is supported here by new evidence involving multiple treatments and different hosts (Figure 3, Figure 4 and Figure 5). In addition, the effect of these treatments on virus concentrations in systemically infected plants supports the conclusion about the defence response as the mode of antiviral activity of *M. croatica* EO. The results of the inhibition of CMVsat (satellite associated *Cucumber mosaic virus*) infection in local and systemic host plants by EO/DC of oil (Figure 3, Figure 4 and Figure 5) leads to the conclusion that the antiphytoviral effect is more affected by the method of EO application and the EO-virus-plant host combinations studied than by the described variations in the oil composition.

### 2.3. Aox Gene Expression

In order to better elucidate the plant defence pathways against viruses at the molecular level, *Arabidopsis thaliana* was used as a plant that is a well-known model system susceptible to CMV infection. The aim was to study the effects of EO/DC of *M. croatica* oil on the expression of one of the important genes in plant defence against pathogens. Changes in the expression of the *Aox* gene in different biotic and/or abiotic stress conditions, such as infection by a pathogen, suggests that AOX belongs to the proteins involved in defence against biotic and abiotic stress [31,32]. The enzyme AOX reduces the formation of new reactive oxygen species (ROS) in the mitochondria and, under stress conditions, can prevent an increased accumulation of ROS molecules [28]. Our experimental results showed that virus infected plants increased *Aox* expression compared to healthy, untreated plants (Figure 6a). Literature reports show that plants exposed to stress conditions, biotic or abiotic, synthesise ROS molecules, and the virus defence response directs the flow of electrons to the enzyme AOX that acts as negative regulator of ROS synthesis [33]. Another study found that increased expression of *Aox* in the Arabidopsis plant is accompanied by an enhanced spread of the virus CMVsat throughout the plant and decreased expression with a slowing of the systemic spread of the virus [34]. We found that plants treated with EO Boj/EO St/β-caryophyllene prior to virus inoculation decreased *Aox* gene expression compared to untreated infected plants (Figure 6b). In accordance with the literature data, it can be assumed that the EO treated plant, rather than the virus, silences the expression of the *Aox* gene, thereby protecting the plant by slowing the spread of CMVsat. Lower expression of the *Aox* gene and increased concentration of mitochondrial ROS molecules protects the plant through the appearance of induced resistance to CMV and a deceleration of the spread of the virus [35]. On the contrary, plants treated with the both essential oil from the locality Bk and caryophyllene oxide showed increased *Aox* gene expression compared to untreated infected plants (Figure 6b). Another study found that AOX induction also occurs in response to other pathogen infections [26,36], and hence may be a general consequence of pathogen infection rather than a specific resistance response to the virus. These results lead to the conclusion that one part of defence response mediated by EO involves a change of expression of the *Aox* gene. This is consistent with Murphy et al. (2004), who suggested that the enzyme AOX could be a regulator of the ROS molecule mediated signalling in plant defence response to viruses [24]. The involvement of EO constituents in plant defence was also described by Hong et al. (2012), who examined the emission of volatile terpenes by *A. thaliana* flowers, which may function in plant–insect interactions [17]. The same authors state that *Arabidopsis* MYC2, a basic helix-loop-helix transcription factor, directly binds to promoters of the sesquiterpene synthase genes, TPS21 and TPS11, and activates their expression. Expression of TPS21 and TPS11 can be induced by the phytohormones GA and JA, and both inductions require MYC2. The induction of TPS21 and TPS11 results in increased emission of sesquiterpene, especially (E)-β-caryophyllene. These results indicate that the production of these sesquiterpenes is integrated with the GA and JA signalling pathways involved in plant defence [17]. Accordingly, it can be concluded that exogenously applied EOs and sesquiterpene constituents of the oil of *M. croatica* are involved in plant defence against CMVsat infection.

The results presented here provide new insight into the antiphytoviral activity of essential oils. Understanding the defence response mechanisms at the molecular level will improve our understanding of the plant defence responses and enhance our knowledge about the antiphytovial potential of essential oils and their use in the prevention of viral plant diseases. The above presents several directions for future research that should aim to clarify the mechanism of antiviral activity of essential oils.

## 3. Materials and Methods

### 3.1. Micromeria Croatica

Randomly selected samples of *Micromeria croatica* (Pers.) Schott were collected from the wild during the blooming period on Mt. Velebit (Croatia) at the localities Bojinac, Bačić kuk and Stupačinovo (abbreviated in text as Boj, Bk and St, respectively). GPS coordinates and altitudes were 44°34′ N, 15°41′ E, elevation 1060 m (Boj); 44°57′ N, 15°10′ E, elevation 1260 m (Bk) and 44°32′ N, 15°10′ E, elevation 1058 m (St). Voucher specimens of herbal material were deposited in the Fran Kušan Herbarium (HFK-HR), Department of Pharmaceutical Botany garden, Faculty of Pharmacy and Biochemistry, University of Zagreb, Zagreb, Croatia.

### 3.2. Essential Oil Isolation

The aboveground parts of several dozen randomly selected plants were harvested from plants on a dry day and mixed to obtain a randomly selected sample. Samples were air-dried for ten days in a well-ventilated room at 60% relative air humidity and room temperature (22 °C), single-layered and protected from direct sunlight. Dried aerial parts (100 g) from each locality were subjected to hydro-distillation for 3 h in a Clevenger type apparatus. Essential oils were dried over anhydrous sodium sulphate and stored at −20 °C for further experiments. The essential oils (EOs) were named according to abbreviation of the locality of collection (EO Boj, EO Bk, EO St). Gas chromatography (GC) analyses were performed on a gas chromatograph (model 3900; Varian Inc., Lake Forest, CA, USA) equipped with a flame ionization detector (FID), mass spectrometer (MS) (model 2100T; Varian Inc.), non-polar capillary column VF-5ms (30 m × 0.25 mm i.d., coating thickness 0.25 μm) and polar CP Wax 52 (30 m × 0.25 mm i.d., coating thickness 0.25 μm; Varian Inc.). Chromatography conditions were as described by Kremer et al. (2012) [18].

### 3.3. Virus and Plant Hosts

Cucumber mosaic virus with associated satellite RNA (CMVsat), previously described by Škorić et al. (1996) [37], was propagated in a systemic host, *Nicotiana megalosiphon* Van Heurck & Müll. Arg. A voucher specimen of this plant virus is deposited at the Faculty of Science, University of Zagreb. Systemically infected leaves were homogenised in phosphate buffer (0.06 mol L^−1^, pH 7.0), briefly centrifuged at 5.000× *g* for 5 min at 4 °C and the supernatant was used for inoculation. Prepared inoculum was diluted with the same buffer to give a suitable number (10–30) of discrete local lesions on test plants. Leaves of the host plants were dusted with silicon carbide (Sigma-Aldrich, St. Louis, MO, USA) prior to inoculation.

Seeds of host plants *Chenopodium quinoa* Willd and *Nicotiana megalosiphon* were sown in trays in a growth chamber (RK-900, Kambič, Semič, Slovenia) at a temperature of 26 °C under a 16:8 h light/dark cycle, with watering as required. Seeds of *Arabidopsis thaliana* (L.) Heynh. ecotype Col-0 were stratified at 4 °C in the dark for two days prior to sowing and after sowing transferred to a growth chamber at a temperature of 23 °C under a 12-h photoperiod. Experimental plants were selected three to four weeks after sowing. Care was taken to ensure that the experimental plants were as uniform in size as possible.

### 3.4. Antiphytoviral Activity Assay on Local Host Plants

#### 3.4.1. Simultaneous Inoculation with Virus

EO Boj, EO Bk, EO St, β-caryophyllene (Sigma-Aldrich, St. Louis, MO, USA) and caryophyllene oxide (Sigma-Aldrich, St. Louis, MO, USA) were added separately to virus inocula at a final concentration adjusted to 500 ppm. Prepared inocula were incubated at room temperature for a short time (0 h), 1, 2 or 3 h. Inoculation on the half-leaves of *C. quinoa* using a sterile glass spatula (0.1 mL amount per leaf half) followed. The corresponding opposite (control) half was rubbed with the control virus inocula to which distilled water was added instead of essential oil or oil compounds. Inoculated leaves were washed thoroughly with distilled water 1 min after inoculation. Ten to fifteen experimental plants (four leaves per plant) were inoculated in each experiment. Local lesions were counted by developing clear and well-visible symptoms (5–6 days after virus inoculation) and the inhibition percentage was calculated by comparing the number of viral lesions on the two leaf halves according to the formula: IP = [(C − T)/C] × 100, where IP = inhibition of local lesions in %, C = mean number of local lesions on the control leaf-half; T = mean number of local lesions on leaf-half inoculated with virus inocula containing essential oil, β-caryophyllene or caryophyllene oxide.

#### 3.4.2. Treatment Prior to Inoculation

EO Boj, EO Bk, EO St, β-caryophyllene and caryophyllene oxide were added separately in distilled water at a final concentration adjusted to 500 ppm. Fresh preparations were sprayed daily on the leaves of *Chenopodium quinoa* plants for one, two or three consecutive days prior to virus inoculation. One hour after the daily treatment, plants were rubbed with virus inocula (0.2 mL per leaf). Control plants were sprayed with distilled water and inoculated with virus inocula. Ten to fifteen experimental plants (four leaves per plant) were inoculated in each experiment. Lesions were counted by developing clear and well-visible symptoms and the inhibition percentage of local lesions number was calculated by comparing the number of viral lesions on the leaves of treated and control plants according to the formula described in Section 3.4.1.

### 3.5. Antiphytoviral Activity Assay on Systemic Host Plants

Systemic host plants *Nicotiana megalosiphon* plants were: (i) inoculated with virus inocula in which EO Boj/EO Bk/EO St/β-caryophyllene/caryophyllene oxide were added at a final concentration adjusted to 500 ppm (simultaneous inoculation); (ii) three consecutive days prior to virus inoculation, plants were sprayed with distilled water in which EO Boj/EO Bk/EO St/β-caryophyllene/caryophyllene oxide were added at a final concentration adjusted to 500 ppm (pretreatment); and (iii) three consecutive days prior and seven consecutive days after virus inoculation, plants were sprayed with distilled water in which EO Boj/EO Bk/EO St/β-caryophyllene or caryophyllene oxide were added at a final concentration adjusted to 500 ppm (continuous treatment). The same number of control plants were rubbed with virus inocula (control for simultaneous inoculation) or were sprayed with distilled water and rubbed with virus inocula (control for pretreatment and continuous treatment). Uninfected plants were used as negative controls. All experimental groups consisted of about 50 plants. The virus was purified from young, uninoculated leaves of systemically infected plants collected at on days 1, 3, 4 and 7 post inoculation (day 0). Leaves were homogenized in chilled 0.01 M Na/K-phosphate buffer (1:1, *w*:*v*), pH 7.5, containing 0.2% DIECA, 0.2% Na_2_SO_3_, and 0.05 M EDTA. Following clarification with ¼ volume chloroform-*n*-butanol (1:1), the virus particles were pelleted by ultracentrifugation (60,000 g, 140 min) and resuspended in 0.01 M homogenization-buffer without additives. The purity and concentration of the virus preparation was measured by a spectrophotometer (BECKMAN DU 530, Beckman Instruments, Inc., Fullerton, CA, USA) according to the formula: C = (A_260_/5) × R, where C = virus concentration, A_260_ = absorption (λ = 260 nm), R = dilution.

The inhibition percentage was calculated by comparing the virus concentration in the essential oil-treated and control groups according to the formula: *Ic* = [(*Cc* − *Ct*)/*Cc*] × 100, where *Ic* = decrease of virus concentration in %, *Cc* = virus concentration in the control group, and *Ct* = virus concentration in the essential oil-treated group. The results are presented as the average from three repeated experiments.

### 3.6. Aox Gene Expression Assays

#### 3.6.1. Treatment of Arabidopsis Thaliana

*Arabidopsis thaliana* plants were divided into eight groups: uninoculated plants, plants rubbed with phosphate buffer, plants inoculated with virus inocula, and plants sprayed with distilled water containing separately EO Boj, EO Bk, EO St, β-caryophyllene or caryophyllene oxide, at a final concentration adjusted to 500 ppm, for three consecutive days prior to virus inoculation. Upper leaves harvested on day 3 post inoculation (day 0) comprised the upper rosette leaves, excluding flower stalks. Three biological replicates from all experimental groups were analysed for *Aox* gene expression. Each group consisted of about 30 plants that were harvested, flash frozen in liquid nitrogen and freeze dried for 24 h in the freeze dryer LIO-5P (Kambič, Semič, Slovenia). Freeze dried tissue was powdered with a mortar and pestle and stored at −80 °C until further analysis.

#### 3.6.2. Quantitative Reverse Transcription-Polymerase Chain Reaction

Total RNA was purified from the collected plant material of *Arabidopsis thaliana* using TRI Reagent Solution (Ambion, Austin, TX, USA) according to the manufacturer’s instructions. For Dnase treatment, 10 μg purified total RNA from each sample was used and processed with 2U TURBO Dnase (Ambion, Austin, TX, USA) according to manufacturer’s instructions. RNA concentration was determined using NanoDrop 2000c (Thermo Fisher Scientific, Waltham, MA, USA). 10 μg total RNA treated with DNase was used for cDNA synthesis and was mixed with oligo d(T)16 and random hexamer primers, both in the final concentration of 0.1 μmol L^−1^. To that mixture, dNTPs (2.5 mmol l^−1^) were added, incubated for 10 min at room temperature and subsequently cooled on ice. For reverse transcription, a master mix (12 μL, Applied Biosystems, Foster City, CA, USA) containing 5 μL of 25 mmol l^−1^ MgCl_2_, 2.5 μL 10× PCR buffer II, Rnase inhibitor in a final concentration of 1 U μL^−1^ and MuLV reverse transcriptase in a final concentration of 2.5 U μL^−1^ was prepared and mixed with total RNA, primers and dNTPs. Each sample was incubated for 45 min at 42 °C and then heated to 99 °C for 5 min.

The cDNA was then used for quantitative Real-Time PCR (qRT-PCR). Expression levels of alternative oxidase 1A—(*Aox*1a) (Assay ID: At02278012_g1) was measured using TaqMan Gene Expression Assays (Applied Biosystems, Foster City, CA, USA) in an optical 96-well plate using a 7300 Real-Time PCR system (Applied Biosystems, Foster City, CA, USA). The procedure was performed according to the manufacturer’s instructions. The average CT values calculated from triplicate samples were used to determine the fold expression relative to the controls. Data were normalized to the expression level of *F-box* (Assay ID: At02200102_s1) and *EF 1α* (Assay ID: At02337969_g1), constitutively expressed reference genes, and relative expression of investigated genes was calculated according to Pfaffl (2004) [38].

### 3.7. Statistical Analysis

All analyses were carried out in triplicate and the data were presented as means ± standard deviations. Statistical significance of the differences between the investigated groups was evaluated using the Student t-test (GraphPad InStat software, San Diego, CA, USA). *p* values < 0.05 were considered statistically significant. Virus concentration is presented as the mean value of two repeated experiments.

Principal component analysis (PCA) and the Pearson correlation coefficient were carried out using the statistical software Statistica 8 (StatSoft, Tulsa, OK, USA). The cluster analysis was performed with the software PRIMER 5.0 PREMIER (Biosoft International, Corina Way, Palo Alto, CA, USA).

## 4. Conclusions

β-caryophyllene and caryophyllene oxide were the main oil components in investigated samples of the essential oil of *Micromeria croatica* from the localities Bojinac, Bačić kuk and Stupačinovo (Mt. Velebit, Croatia). An antiphytoviral assay showed that simultaneous inoculation of CMVsat with essential oil or the dominant components of the oil, or treatment of plants with essential oil or the dominant components of the oil prior to virus inoculation resulted in a reduction of the number of lesions on the leaves of local host plants, and a reduction of the virus concentration in systemically infected plants. Antiviral activity of the essential oil was correlated with the antiviral activity of the main oil component, demonstrating the role of β-caryophyllene and caryophyllene oxide in the activity of the essential oil in plant defence against viruses. At least one part of the defence response mediated by essential oil involves a change in the level of expression of the alternative oxidase (*Aox*) gene.

## Figures and Tables

**Figure 1 molecules-24-01342-f001:**
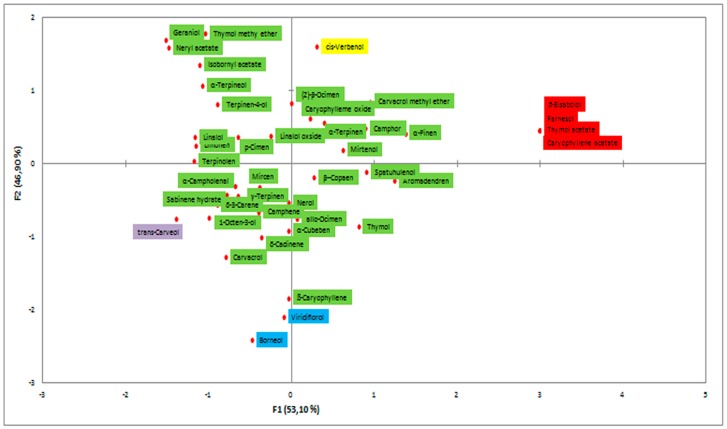
PCA of essential oil (EO) components in *Micromeria croatica* from the localities Bojinac (Boj), Bačić kuk (Bk) and Stupačinovo (St). 
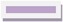
 EO components identified in *M. croatica* from Boj and Bk; 
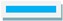
 EO components identified in *M. croatica* from Boj and St; 
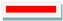
 EO components identified in *M. croatica* from St; 
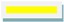
 EO components identified in *M. croatica* from St and Bk.

**Figure 2 molecules-24-01342-f002:**
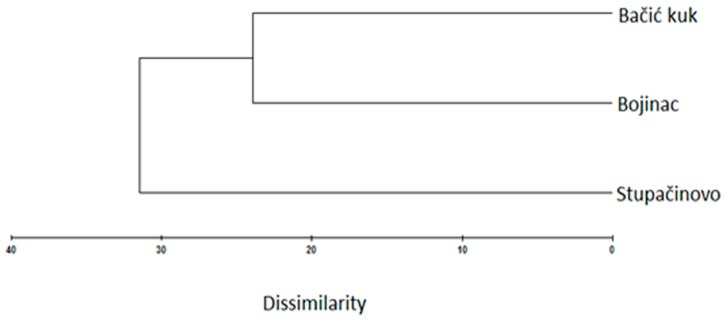
Complete linkage cluster of the essential oil of *Micromeria croatica* from the localities Bojinac, Bačić kuk and Supačinovo based on Bray Curtis dissimilarity.

**Figure 3 molecules-24-01342-f003:**
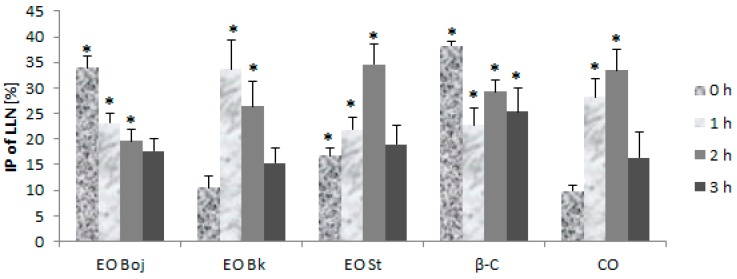
Inhibition percentage (IP) of local lesion number (LLN) on the half-leaves of *Chenopodium quinoa* plants inoculated with *Cucumber mosaic virus* containing associated satellite RNA (CMVsat). Prior to inoculation, the virus was incubated for a short time (0 h), 1 h, 2 h or 3 h with essential oil (EO) of *Micromeria croatica* from the localities Bojinac (Boj), Bačić kuk (Bk) and Stupačinovo (St), β-caryophyllene (β-C) and caryophyllene oxide (CO). Control half-leaves were inoculated with virus inocula without EO or oil components. Ten to fifteen experimental plants (four leaves per plant) were inoculated in each group. Values are expressed as the mean value + SD calculated based on the three independent evaluations (n = 3). The asterisk (*) denotes significant inhibition of LLN compared to the control (*p* < 0.05).

**Figure 4 molecules-24-01342-f004:**
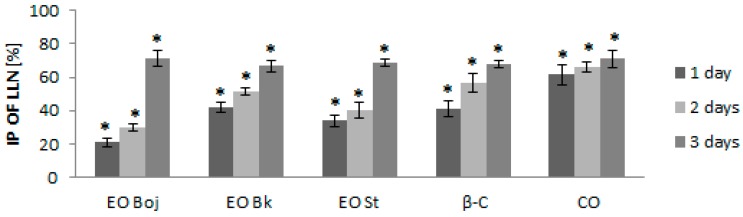
Inhibition percentage (IP) of local lesion number (LLN) on the leaves of *Chenopodium quinoa* plants inoculated with *Cucumber mosaic virus* containing associated satellite RNA (CMVsat). Prior to inoculation, plants were sprayed with essential oil (EO) of *Micromeria croatica* from the localities Bojinac (Boj), Bačić kuk (Bk) and Stupačinovo (St), β-caryophyllene (β-C) and caryophyllene oxide (CO) for 1, 2 or 3 consecutive days. Control plants were inoculated with virus inocula without EO or oil components. Ten to fifteen experimental plants (four leaves per plant) were inoculated in each group. Values are expressed as the mean value ± SD calculated based on three independent evaluations (n = 3). The asterisk (*) denotes significant inhibition of local lesions number compared to the control (*p* < 0.05).

**Figure 5 molecules-24-01342-f005:**
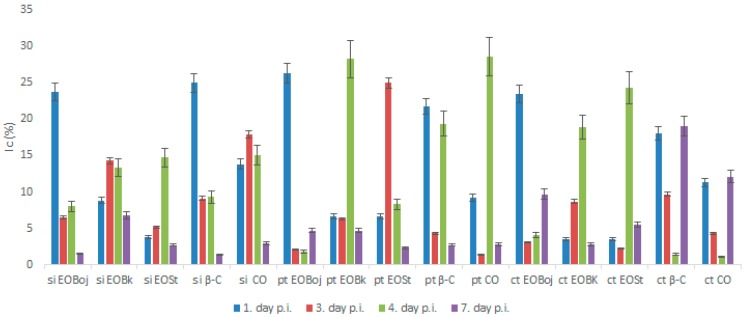
Inhibition of CMVsat (satellite associated *Cucumber mosaic virus*) concentration (Ic) in *Nicotiana megalosiphon* plants following: Simultaneous inoculation (si) of virus inocula with essential oil (EO) of *Micromeria croatica* from the localities Bojinac (Boj), Bačić kuk (Bk) and Stupačinovo (St) or with β-caryophyllene (β-C) and caryophyllene oxide (CO); treatment prior to inoculation (pt) with EO of *M. croatica* from the localities Boj, Bk and St or with β-C and CO; continuous treatment (ct) prior and after inoculation with EO of *M. croatica* from the localities Boj, Bk and St or with β-C and CO, all related to CMVsat concentration in control plants. All experimental groups consisted of about 50 plants. Leaves were harvested on days 1, 3, 4 and 7, post inoculation (p.i.). Values are expressed as the mean value ± SD calculated based on three independent experiments.

**Figure 6 molecules-24-01342-f006:**
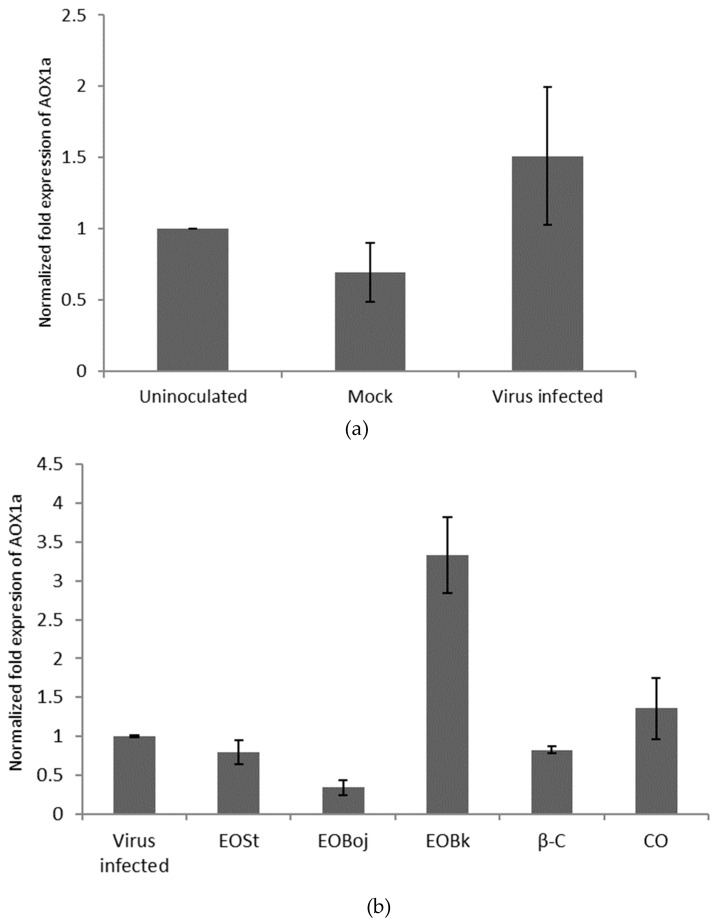
Relative expression of the alternative oxidase (*Aox1a*) gene in upper leaves of CMVsat infected *Arabidopsis thaliana* plants determined with qRT-PCR. Plants were: uninoculated, buffer-treated (Mock); infected with *Cucumber mosaic virus* containing associated satellite RNA (Virus infected); sprayed with essential oil (EO) of *Micromeria croatica* from the localities Bojinac (Boj), Bačić kuk (Bk) and Stupačinovo (St) or with β-caryophyllene (β-C) and caryophyllene oxide (CO) three consecutive days prior to inoculation of the virus. Error bars represent standard deviation calculated from three biological replicates. Normalisation factors were calculated as the geometric mean of the expression levels of the reference genes (F-box and EF 1α). Uninoculated (**a**) or virus-infected (**b**) sample was used as the calibrator (=1). Leaves were harvested on the third day after inoculation. The asterisk (*) denotes a significant change in the expression of the *Aox1a* compared to the control (*p* ˂ 0.05).

**Table 1 molecules-24-01342-t001:** Phytochemical composition (%), identification and major groups of chemical components of the essential oil of *Micromeria croatica* (Pers.) Schott from the localities Bojinac (Boj), Bačić kuk (Bk) and Stupačinovo (St) *.

			Sample (Yield in %) ^c^		
Component	RI ^a^	RI ^b^	*Micromeria croatica*		
			Boj ± SD (1.3)	Bk ± SD (0.7)	St ± SD (0.6)
***Monoterpene hydrocarbons***			**21.8**	**24.5**	**14.0**
*α*-Thujene	924	1035	-	0.9 ± 0.03	-
*α*-Pinene **	938	1032	0.5 ± 0,15	1.1 ± 0.01	1.0 ± 0.01
Camphene **	962	1059	0.9 ± 0.01	0.4 ± 0.01	0.6 ± 0.01
1-Octen-3-ol	974	1452	1.2 ± 0.05	1.1 ± 0.01	0.3 ± 0.01
Myrcene	992	1174	2.4 ± 0.01	1.4 ± 0.01	0.8 ± 0.01
*δ*-3-Carene	1008	1153	2.5 ± 0.01	1.8 ± 0.01	0.6 ± 0.1
*α*-Terpinene	1016	1179	2.7 ± 0.01	4.2 ± 0.05	5.1 ± 0.01
*p-*Cymene	1021	1268	1.5 ± 0.05	3.3 ± 0.01	1.3 ± 0.06
Limonene	1032	1204	3.9 ± 0.06	5.9 ± 0.01	0.8 ± 0.01
(*Z*)-β-Ocimene **	1032	1218	0.4 ± 0.01	1.6 ± 0.01	0.9 ± 0.01
*γ-*Terpinene	1057	1255	3.5 ± 0.03	2.6 ± 0.01	1.6 ± 0.01
*allo-*Ocimene	1128	1351	2.3 ± 0.01	0.9 ± 0.01	1.0 ± 0.03
***Oxygenated monoterpenes***			**25.8**	**41.3**	**27.6**
Linalool oxide **	991	-	0.9 ± 0.01	0.8 ± 0.01	0.5 ± 0.01
Sabinene hydrate	1065	1474	1.1 ± 0.04	0.7 ± 0.01	0.4 ± 0.01
Terpinolene	1085	1286	1.6 ± 0.03	2.4 ± 0.1	0.4 ± 0.07
Linalool	1097	1548	3.5 ± 0.03	6.4 ± 0.01	1.1 ± 0.01
*α*-Campholenal	1122	-	1.8 ± 0.01	1.2 ± 0.01	0.7 ± 0.02
*cis-*Verbenol	1137	-	0.2 ± 0.01	0.7 ± 0.03	0.5 ± 0.01
Camphor	1143	1499	1.9 ± 0.01	3.1 ± 0.01	6.8 ± 0.01
Borneol	1165	1719	2.2 ± 0.03	-	0.2 ± 0.01
Terpinen-4-ol	1174	1611	1 ± 0.01	2.5 ± 0.01	0.7 ± 0.01
*α*-Terpineol	1186	1646	1.1 ± 0.01	2.1 ± 0.01	0.5 ± 0.1
Myrtenol	1194	1804	1.4 ± 0.03	2.5 ± 0.01	3.6 ± 0.03
trans-Carveol	1216	-	1 ± 0.01	1.3 ± 0.01	-
Nerol	1227	1808	2 ± 0.04	0.6 ± 0.01	1.2 ± 0.01
Thymol methyl ether	1230	1604	0.5 ± 0.01	3.3 ± 0.01	0.8 ± 0.01
Carvacol methyl ether	1241	1614	0.2 ± 0.01	1.1 ± 0.01	2.1 ± 0.01
Geraniol	1249	1857	0.8 ± 0.01	5.1 ± 0.01	0.6 ± 0.01
Isobornyl acetate	1283	-	0.4 ± 0.01	1.7 ± 0.01	0.2 ± 0.04
Thymol	1290	-	1.2 ± 0.01	0.2 ± 0.01	1.1 ± 0.01
Carvacrol	1298	2239	2.2 ± 0.01	0.7 ± 0.01	0.4 ± 0.03
Thymol acetate	1349	-	-	-	5.4 ± 0.01
Neryl acetate	1358	1692	0.8 ± 0.1	4.9 ± 0.01	0.4 ± 0.03
***Sesquiterpene hydrocarbons***			**31.4**	**5.0**	**16.3**
*α*-Cubebene	1345	-	3.5 ± 0.07	1.4 ± 0.01	2.3 ± 0.01
β-Caryophyllene **	1417	1585	25.2 ± 0.03	1.8 ± 0.01	10.2 ± 0.01
β-Copaene	1430	1457	1 ± 0.01	0.7 ± 0.01	1.4 ± 0.01
Aromadendrene	1439	-	0.7 ± 0.02	0.3 ± 0.01	1.9 ± 0.1
*δ*-Cadinene	1517	1745	1 ± 0.01	0.8 ± 0.01	0.5 ± 0.01
***Oxygenated sesquiterpenes***			**15.0**	**21.6**	**32.9**
Sphatulenol	1578	2144	1.6 ± 0.04	0.5 ± 0.01	2.2 ± 0.01
Caryophyllene oxide	1582	1927	10.1 ± 0.08	21.1 ± 0.01	20.2 ± 0.01
Viridiflorol	1592	-	3.3 ± 0.01	-	1.1 ± 0.01
β-Bisabolol	1674	2166	-	-	3.0 ± 0.01
Caryophyllene acetate **	1701	-	-	-	4.9 ± 0.1
Farnesol	1742	-	-	-	1.5 ± 0.01
***Total identified (%)***			**94.0**	**92.4**	**90.8**

Retention indices (RI) were determined relative to a series of *n*-alkanes (C_8_–C_40_) on capillary columns VF5-ms (RI ^a^) and CP Wax 52 (RI ^b^); ^c^ average value from two columns, where possible; * identification method: RI, comparison of RIs with those listed in a homemade library, reported in the literature [30], and/or authentic samples; comparison of mass spectra with those in mass spectral libraries NIST02 and Wiley 9; ** co-injection with reference compounds; SD, standard deviation.

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
