# Peer review of "Inhibition of Satellite RNA Associated Cucumber Mosaic Virus Infection by Essential Oil of Micromeria croatica (Pers.) Schott"

_molecules, 2019, doi:10.3390/molecules24071342_

Round 1

Reviewer 1 Report

The authors have done a good job in presenting and discussing the data. Please add standard deviations in Table 1. The manuscript is well written. Minor spell checks needed.

Author Response

Response to Reviewer 1 Comments

Point 1: The authors have done a good job in presenting and discussing the data. Please add standard deviations in Table 1. The manuscript is well written. Minor spell checks needed.

Response 1: Thank you for your comment. Standard deviations are added in Table 1. We  have made spell check and minor corrections are marked in the text.

Reviewer 2 Report

Generally speaking, this paper is well written, and the major group of chemical components of the essential oil of Micromeria croatica are very clear.

However, there are several points that should be made them clear:

In Fig4, only n =3 , what is the statistical method which can analyze the significant ihibition of local lesions number compared to the control? (n= ; p<0.05)

Fig5 is unclear and should be corrected.

In Fig3-6, the Y-axis background line should be deleted.

In Fig3 and Fig4, the sample sizes are small (n=3) and should not adapt the student t-test.  They should be analyzed by nonparametric Statistics. 

Author Response

Response to Reviewer 2 Comments

Point 1: Generally speaking, this paper is well written, and the major group of chemical components of the essential oil of Micromeria croatica are very clear. However, there are several points that should be made them clear: In Fig4, only n =3 , what is the statistical method which can analyze the significant ihibition of local lesions number compared to the control? (n=3 ; p<0.05).

Response 1: Statistical significance of the differences between the investigated groups was evaluated using the Student t-test. P values ˂ 0.05 were considered statistically significant. As described in chapter 3.4.1. Ten to fifteen experimental plants (four leaves per plant) were inoculated in each group. All analyses were carried out in triplicate (n = 3) and the data were presented as means ± standard deviations. Thus, amount of our experimental results was appropriate for applied statistical analysis. Explanation is added below Fig 4.

Point 2: Fig5 is unclear and should be corrected.

Response 2:  New Figure 5 is given. Figure 5 caption is corrected as follows:

Figure 5. Inhibition of CMVsat concentration (Ic) in Nicotiana megalosiphon plants following: simultaneous inoculation (si) of virus inocula with essential oil (EO) of Micromeria croatica from the localities Bojinac (Boj), Bačić kuk (Bk) and Stupačinovo (St) or with β-caryophyllene (β-C) and caryophyllene oxide (CO); treatment prior to inoculation (pt) with EO of M. croatica from the localities Boj, Bk and St or with  β-C and CO; continuous treatment (ct) prior and after inoculation with EO of M. croatica from the localities Boj, Bk and St or with β-C and CO, all related to CMVsat concentration in control plants. All experimental groups consisted of about 50 plants. Leaves were harvested on days 1, 3, 4 and 7 post inoculation (p.i.). Values are expressed as the mean value value ± SD calculated based on three independent experiments.

Point 3: In Fig3-6, the Y-axis background line should be deleted

Response 3: Y-axis background line is deleted in Fig 3-6

Point 4: In Fig3 and Fig4, the sample sizes are small (n = 3) and should not adapt the student t-test. They should be analysed by nonparametric Statistics

Response 4:    As described in chapter 3.4.1. Ten to fifteen experimental plants (four leaves per plant) were inoculated in each group. All analyses were carried out in triplicate (n = 3) and the data were presented as means ± standard deviations. Thus, amount of our experimental results was appropriate for applied statistical analysis. Explanation is added below Fig 3 and Fig 4.

Reviewer 3 Report

It is interesting but results are different in EO species. several questions.

L.176, Fig3. and L.425. Why did you use 500 ppm. 500 ppm may be high? Is there any effect on plant, safe? How much the highest concentration of oil without effect on plants? Describe.

In Fig 3, Increasing time, effect was decreased in Boj, but increased in St. Describe the reason. Why effect of oil on antiviral effect was different in time with species.  Boj decreased at longer incubation time, while St increased in longer incubation.

Fig 5, is there any significance between the case? It seems no rule. What is the evidence that you can get from those data? Show SD.

Fig 6. Expression of AOX1a was decreased by St Boj and Beta-C but Bk increased. Why?

L.473,  you describes “..by EO involves a change in …AOX..”, but sometimes increased, and sometimes decreased. Why you can say this conclusion.

Dose Sesquiterpenes rich plant have same effect? If so describe.

Author Response

Response to Reviewer 3 Comments

Point 1: L.176, Fig3. and L.425. Why did you use 500 ppm. 500 ppm may be high? Is there any effect on plant, safe? How much the highest concentration of oil without effect on plants? Describe.

Response 1: Our preliminary experiments showed that 500 ppm concentration of Micromeria croatica essential oil is safe, without any toxic effect on experimental plants. Lower concentrations of oil were much less effective in reduction of virus infection. Higher concentrations did not improve antiphytoviral effect. Thus, we used 500 ppm as experimental essential oil concentration.

Point 2: In Fig 3, Increasing time, effect was decreased in Boj, but increased in St. Describe the reason. Why effect of oil on antiviral effect was different in time with species.  Boj decreased at longer incubation time, while St increased in longer incubation.

Response 2: As we explained in the Introduction section, “…the aim of this study was to provide answers to the following questions: …. ii) do different application methods of EO/DC affect antiviral activity in local host plants; iii) does an extended incubation time of the virus with EO/DC reflect on the antiviral activity in local host plants… v) does the prolongation of treatment with EO/DC prior to inoculation reflect on the antiviral activity in local host plants…vii) is there a correlation between the antiviral activity of the EO and the antiviral activity of the DC of oil”

When inoculation was carried out after a short incubation period (0 h), EO Boj and β-caryophyllene were most effective in reducing the number of local lesions compared to the control (Fig. 3). EO St and caryophyllene oxide were most effective during the experiment with two-hour preincubation with the virus in comparison to the control. Based on the experimental results, a positive correlation of the antiviral activity of EO Boj with β-caryophyllene is established. Antiviral activity of EO Bk and EO St positively correlated with the antiviral activity of caryophyllene oxide. The antiphytoviral assay showed that activity of the essential oil correlated with the antiviral activity of the main oil component, demonstrating the role of β-caryophyllene and caryophyllene oxide in the activity of the essential oil in plant defence against viruses. Preincubation of viral inoculum with essential oil/dominant components (DC) of the oil that lasted more than two hours did not enhance the antiphytoviral effect (Fig. 3). Due to the volatility of EOs, the concentration of active compounds was likely reduced. Since both the PCA and cluster analysis showed that composition of the oil from locality St was specific (Fig. 1, Fig. 2), it is possible that this slightly different oil composition affects its antiviral activity. It is possible that aside from the main components, other components in this oil or their synergistic effect contributed to the antiviral activity of oil from St.

Point 3: Fig 5, is there any significance between the case? It seems no rule. What is the evidence that you can get from those data? Show SD.

Response 3: Overall intention was to correlate the activity of the oil and activity of its main components and to examine how methods of application and differences in oil composition reflect on the antiphytoviral activity of the oil. Figure 5 shows virus concentrations in leaves of EO-treated and untreated Nicotiana megalosiphon plants. Simultaneous inoculation of systemic host plants established a positive correlation of the antiviral effect of EO Boj with β-caryophyllene. Pretreatment of systemic host plants was more successful in reducing virus concentrations than simultaneous inoculation in all experimental groups and data established a positive correlation of the antiviral activity of EO Bk with caryophyllene oxide. Continuous treatment of systemic host plants prior to and after inoculation did not improve the antiphytoviral effect over treatment prior to inoculation. Based on the overall results of inhibition of viral infection by EO, it was concluded that the antiphytoviral activity of M. croatica EO positively correlated with the activity of the main oil components. Observed discrepancies from this rule regarding systemic host plants and EO from St locality are the answer to the question of whether antiviral activity is constant given the differences in oil composition.  Since both the PCA and cluster analysis showed specificity of the oil from locality St (Fig. 1, Fig. 2), it is possible that this difference affects antiviral activity of oil from St. It is possible that aside from the main components, other components or their synergistic effect contributed to the antiviral activity of oil from St. Standard deviations are added to Fig 5.

Point 6 and 7: Fig 6. Expression of AOX1a was decreased by St Boj and Beta-C but Bk increased. Why? L.473,  you describes “..by EO involves a change in …AOX..”, but sometimes increased, and sometimes decreased. Why you can say this conclusion

Response 6-7: We included new methodological approaches in this field with the aim of correlating EO composition/method of application/antiviral activity/gene expression. The conclusions were reached based on the obtained results and in accordance with the literature data. Although the connection of plant terpenes and other constituents of essential oils in the induction of plant defences has been described in the literature, as far as we know, this was the first investigation of the effects of exogenously applied essential oil and oil constituents on defence gene expression. Although our results lead to the conclusion that at least one part of the defence response mediated by EO involves a change of expression of the Aox gene, further research and testing of additional genes will provide new insights and clarify the mechanism of the antiviral activity of essential oils.

We found that plants treated with EO Boj/ EO St/ β-caryophyllene prior to virus inoculation decreased Aox gene expression compared to untreated infected plants (Fig. 6b). In accordance with the literature data, it can be assumed that the EO treated plant, rather than the virus, silences the expression of the Aox gene, thereby protecting the plant by slowing the spread of CMVsat. Lower expression of the Aox gene and increased concentration of mitochondrial ROS molecules protects the plant through the appearance of induced resistance to CMV and a deceleration of the spread of the virus [35]. Plants treated with the both essential oil from the locality Bk and caryophyllene oxide showed increased Aox gene expression compared to untreated infected plants (Fig. 6b). Another study found that AOX induction also occurs in response to other pathogen infections  [26,36], and hence may be a general consequence of pathogen infection rather than a specific resistance response to the virus. These results lead to the conclusion that one part of defence response mediated by EO involves a change of expression of the Aox gene.

Point 8: Dose Sesquiterpenes rich plant have same effect? If so describe.

Response 8: Regarding phytopathogenic viruses, various substances of natural and synthetic origin have been assessed for their antiphytoviral activity. So far, limited number of scientific publications have revealed the antiphytoviral activity of essential oils. Although limited, the results indicate that these plant metabolites can trigger a response to viral infection. Qualitative differences in essential oil composition of different plant species do not exhibit a uniform response that would explain the mechanism of antiphytoviral activity. Studies dealing with antiphytoviral activity of essential oils have led to the conclusion that sesquiterpene-rich oils can reduce viral infection in local host plants (Dunkić et al. 2010; 2011; Bezić et al. 2011; Vuko et al. 2012). GC-MS analysis highlighted sesquiterpenes as main components in the oil composition of Micromeria croatica, indicating its possible antiphytoviral activity. Thus, we focused on this promising biological activity of Micromeria essential oil that is unsufficiently explored and consequently opens new area of research that could help in a control of diseases caused by plant viruses.